# Deep Adaptive Cross Domain Learning for Continuous Pressure Wave Signal Recovery

## Abstract

Mud pulse telemetry (MPT) enables real-time downhole data transmission by generating continuous pressure wave signals in drilling fluid, supporting measurement and control during drilling. However, the signals are vulnerable to noise from the downhole environment, mud channel, and surface equipment, causing attenuation, distortion, and phase errors that hinder accurate reconstruction. Recent deep learning methods show promise but typically require large labeled datasets, which are costly and difficult to obtain in the field. To address this issue, this paper proposes the Deep Adaptive Cross Domain Learning Network (**DACDL**), a framework featuring a novel noise adaptation mechanism that transitions from model-level to input-level adaptation. Our approach introduces three core innovations: (1) An Episodic Learning Framework (**EL-Framework**) that simulates domain shift by alternately learning from simulated (gaussian) and real-world (real-world gaussian, pump or oilfield) domains, enhancing few-shot adaptation under label scarcity; (2) A lightweight Adaptive Noise Learning Block (**ANL-Block**) that introduces sample-specific perturbations to align target input noise distributions with the source domain, correcting amplitude attenuation and phase distortion, thus alleviating generalization collapse due to unseen noise characteristics; (3) A Frequency-aware Adversarial Alignment Block (**FAA-Block**) that aligns spectral characteristics between source and target domains, effectively mitigating phase errors and frequency-domain mismatches to enhance cross-domain signal reconstruction. Moreover, the proposed ANL-Block is model-agnostic and can be plug-and-play into most existing methods. Experimental results on three collected datasets demonstrate the effectiveness of DACDL in practical field scenarios and highlight the model-agnostic adaptability of the ANL-Block.

## 1 Introduction

In Mud Pulse Telemetry (MPT) systems (Fig. 1 (a)), pressure signals generated near the drill bit must propagate through an extended, noise-susceptible fluid column to reach surface receivers Jia et al. (2023); Berro & Reich (2019). During propagation, these signals undergo substantial degradation due to three primary factors: (1) attenuation and dispersion induced by the viscous mud medium, (2) multipath propagation resulting from reflections at pipe joints, and (3) contamination by both periodic and stochastic noise sources Li & Xu (2023). The mud pump constitutes a dominant interference source, generating intense periodic noise that frequently spectrally overlaps with telemetry signals in both temporal and frequency domains. Concurrently, random disturbances arising from fluid turbulence and mechanical vibrations further degrade the signal-to-noise ratio (SNR) Shao et al. (2017). These impairments accumulate proportionally with transmission distance, yielding highly distorted and severely attenuated received signals. Consequently, accurate signal reconstruction proves particularly challenging in high-speed continuous pressure wave MPT systems due to pronounced spectral interference within compound noise environments.

While deep learning techniques Chu et al. (2024); Qi et al. (2024); Zhang et al. (2024) have substantially mitigated this limitation, their effectiveness remains contingent upon the availability of adequate labeled data. However, in practical drilling operations, acquiring pre-labeled datasets that accurately characterize downhole mud channel properties entails prohibitive financial and material expenditures, rendering this approach fundamentally impractical.

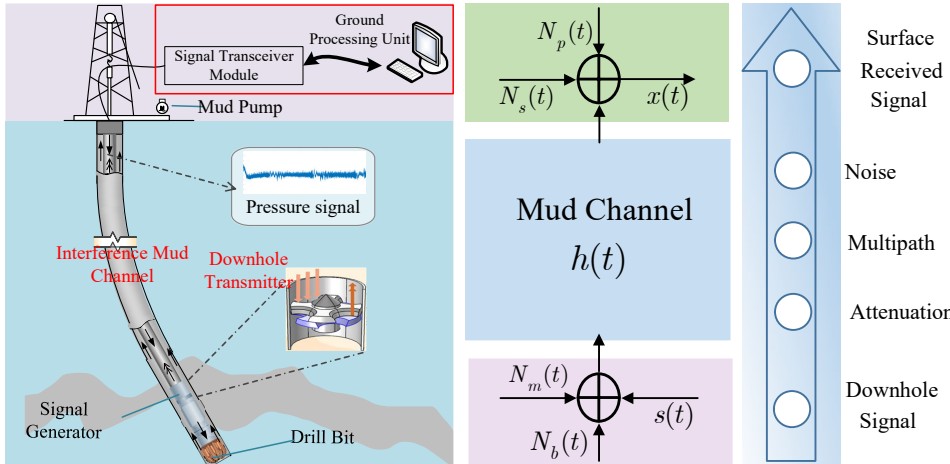

(a) Illustration of the MPT system.    (b) Noise model of continuous pressure wave.

Figure 1: (a) MPT system for downhole drilling. The system is composed of three main modules: 1) a downhole transmitter (signal generator and modulation encoder, located near the drill bit), 2) the mud channel (drilling fluid inside the drill string, labeled as "Interference Mud Channel"), and 3) the ground receiver, which is highlighted within the red box and includes the pressure sensor, signal transceiver module, and ground processing unit. (b) Noise model of pressure wave signal.

To address this issue, we adopt episodic learning in a cross-domain setting: computer-generated simulation data serve as the source domain, while lab data (collected on an indoor hydraulic circulation setup) and field data (acquired in real drilling operations) serve as the target domain. The proposed network architecture is designed to support alternating learning between the source and target domains, thereby facilitating smoother adaptation from the source distribution to the target distribution.

In this context, the **D**eep **A**daptive **C**ross **D**omain **L**earning Network (**DACDL**) is proposed, which introduces a novel noise adaptation mechanism that transitions from model-level to input-level adaptation. The framework comprises three principal components: (1) The **E**pisode **L**earning Framework (**EL-Framework**) offers a platform for alternately learning knowledge from the source domain and the target domain. (2) The **A**daptive **N**oise **L**earning Block (**ANL-Block**) introduces sample-specific perturbations to mitigate generalization collapse caused by previously unseen noise characteristics. (3) The **F**requency-aware **A**dversarial **A**lignment Block (**FAA-Block**) employs a domain discriminator to align the frequency distributions of source and target signals, thereby mitigating domain shift under real-world noise conditions.

The main contributions are summarized as follows:

- This present a novel method named DACDL that realizess the joint learning of source domain and target domain knowledge.

- The ANL-Block is a model-agnostic module designed for seamless integration into diverse deep learning architectures, enabling robust correction of amplitude attenuation and waveform distortion across domains.

- The FAA-Block promotes cross-domain generalization by enforcing spectral invariance through adversarial training in the fourier domain to enhance cross-domain signal reconstruction.

- Our method has been evaluated across 3 datasets. The results have evaluated the efficiency and the model-agnostic functionality of our method.

## 2 RELEVANT CONCEPTIONS

To better explain the proposed DACDL for mud pulse signal denoising, several relevant concepts are introduced in the rest of this section.

### 2.1 DOMAIN SHIFT IN SIGNAL RECOVERY

Since labeled data are scarce, signal recovery models are commonly trained on simulated data with Gaussian noise, which can approximate some real-world conditions. However, real mud telemetry signals often contain complex noise types such as periodic pump noise and impulsive disturbances, which are difficult to reproduce in simulation and result in domain discrepancies in both temporal and spectral characteristics. This domain gap limits the generalization of models trained only on simulated data.

Inspired by progress in image processing, the cross-domain learning is introduced to pressure wave signal denoising and recast the task as a domain generalization problem.

### 2.2 CROSS DOMAIN FEW SHOT LEARNING

Under the cross-domain FSL setting Chen et al. (2024a); Kang et al. (2025); Chen et al. (2024b), two domain data sets are given: source domain data set $\mathcal{D}_S$ with Gaussian noise distribution and target domain data set with $\mathcal{D}_T$ Pump noise distribution. According to whether the data are labeled, target domain data set can further split into two parts: few-shot data set $\mathcal{D}_f$ with labeled data and test data set $\mathcal{D}_t$ with unlabeled data, and $\mathcal{D}_f \cup \mathcal{D}_t = \mathcal{D}_T$. The few-shot data set gets its name since it is relatively small compared with the testing data set.

In DACDL, labeled target-domain samples are obtained by applying a sliding window of length 500 to a known $\mathcal{M}$ sequence. The $\mathcal{M}$ sequence is a known pressure wave signal transmitted periodically from downhole.

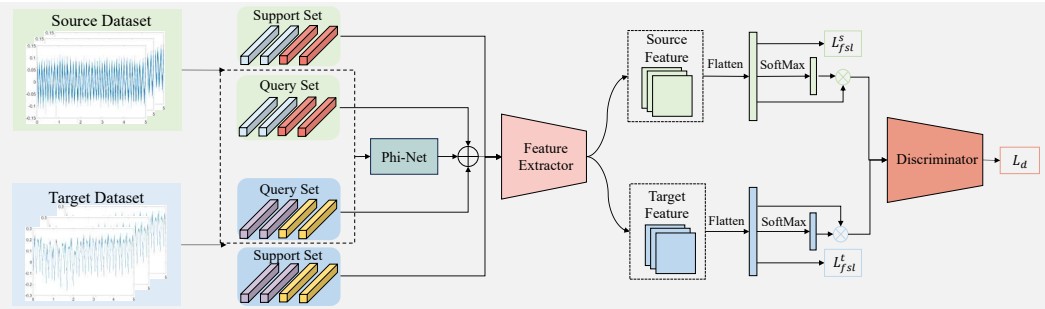

Figure 2: The overall architecture of the proposed Deep Adaptive Cross Domain Learning Network (**DACDL**).

## 3 PROBLEM STATEMENT

As shown in Fig. 1 (b), in mud pulse telemetry systems, the received pressure signal is severely degraded by multipath channel effects and various noise sources. These effects not only attenuate the amplitude but also introduce significant phase distortion, severely complicating the recovery of weak signals.

The observed pressure wave signal can be modeled as:

$$y(t) = s(t) * h(t) + N(t), \tag{1}$$

where $s(t)$ is the transmitted phase-modulated signal, $h(t)$ is the channel impulse response, and $N(t)$ is the aggregate noise.

The aggregate noise can be further expressed as Qu et al. (2021); Chen et al. (2021):

$$N(t) = [N_b(t) + N_m(t)] * h(t) + N_s(t) + N_p(t), \tag{2}$$

where $N_s(t)$ (surface noise) and $N_b(t)$ (downhole noise) are modeled as white noise, $N_p(t)$ (pump noise) contains both periodic pulsating and white noise components, and $N_m(t)$ (mechanical noise) comprises non-periodic pulsating noise together with white noise.

In the frequency domain, the observed signal is:

$$X(u) = S(u)H(u) = |S(u)||H(u)| \, e^{j(\phi_s(u)+\phi_h(u))}, \tag{3}$$

where $\phi_h(u)$ denotes the channel-induced phase distortion.

The goal is to recover the original phase-modulated signal $s(t)$ from the degraded observation $y(t)$.

# 4 METHODOLOGY

## 4.1 OVERVIEW

To robustly denoise pressure-wave signals under real-world domain shifts, we propose DADCL, a domain-adaptive meta-learning framework inspired by few-shot image classification Li et al. (2024; 2025). As outlined in Fig. 2, DADCL pretrains a denoiser on simulated pressure wave signals and meta-adapts it using only a small set of real noisy samples. Because the simulator injects only Gaussian noise, omitting key field characteristics such as phase distortion and waveform overlap, our meta-learning procedure bridges this sim-to-real gap.

## 4.2 WAVELET BASED BACKBONE

In prior work, we introduced WaveU-Net, a wavelet-enhanced U-Net that achieved strong results in signal restoration. Owing to its effectiveness, we adopt it here as the backbone feature extractor and briefly summarize its design. The network augments U-Net with a Learnable Wavelet Denoising Network (LWDNet) inserted between encoder and decoder. LWDNet performs a learnable multi-level wavelet transform on intermediate features, enabling time–frequency analysis that improves denoising and feature extraction.

## 4.3 EPISODE LEARNING FRAMEWORK

Real (target) and simulated (source) signals differ markedly in acquisition settings, instrumentation, and noise, producing a domain gap that degrades "train-on-source, test-on-target" generalization. Our module aligns the spectral features of model outputs across domains to support robust adaptation in complex real-world settings.

**Support and Query Sets**  To bridge the domain gap between the simulated (source) and real (target) signal distributions, this paper adopts an episodic training strategy. At each adaptation step, a meta-task is constructed by sampling support-query pairs from either the source or target domain. Specifically, we form a support set with $\mathcal{K}$ paired clean and noisy signals in the source domain or the target domain, and a query set with a small number of test signals from the same domain.

**Feature Update**  The feature extractor is implemented via the WaveU-Net denoising backbone, followed by a temporal projection head to obtain latent prototypes Li et al. (2024). The denoising loss between the predicted query output and ground truth is computed using an $\mathcal{L}_1$ loss:

$$\mathcal{L}_{fsl} = \|\hat{x} - x\|^2 \tag{4}$$

where $\hat{x}$ denotes the query output and $x$ denotes the query ground truth, respectively.

Notably, the few shot learning (FSL) procedure is identical for both the source and target domains, with the only difference being the number of support and query samples available in each case.

## 4.4 ADAPTIVE NOISE LEARNING BLOCK

The core component of the **A**daptive **N**oise **L**earning Block (**ANL**-Blcok) is Phi-Net, a learnable perturbation module that injects task-specific noise into the input. This allows the pre-trained model to interpret the modified input as being drawn from the distribution of previously encountered noise, thereby facilitating effective adaptation.

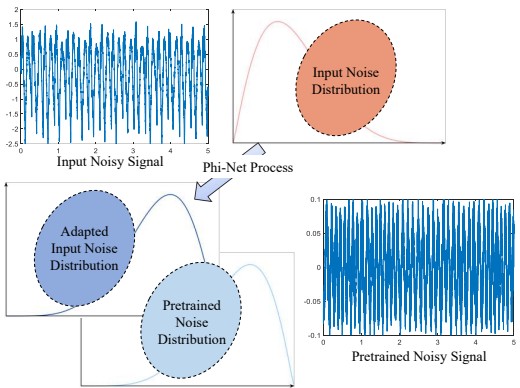

Figure 3: Overview of the ANL-Block.

**Process of Phi-Net** We first formulate a new target domain noise as deviation from the source noise distribution:

$$N^t = N^s + \epsilon^{s\to t} \tag{5}$$

where $\epsilon^{s\to t}$ represents how much $N^t$ deviates from an arbitrary noise $N^s$ sampled from $\mathcal{D}^s$ (source domain); $N^t$ denotes the new target domain noise. Thus, a new noisy input signal in target domain can be seen as follows:

$$y^t = s^t + N^t \tag{6}$$
$$= s^t + N^s + \epsilon^{s\to t}. \tag{7}$$

According to Eq. 6 and Eq. 7, it can be observed that we can mitigate the misalignment issues if we can adapt a given noisy signal $y^t$ to its translated counterpart noisy signal $y^{s\to t}$ with source noise $N^s \sim \mathcal{D}^s$, by removing the deviations $\epsilon^{s\to t}$ as:

$$y^{t\to s} := s^t + N^s \tag{8}$$
$$= s^t + N^s + \epsilon^{s\to t} - \epsilon^{s\to t} \tag{9}$$
$$= y^t - \epsilon^{s\to t} \tag{10}$$

Consequently, our objective is to estimate a deviation offset $-\epsilon^{s\to t}$ that can be applied to a given noisy signal $y^t$, thereby aligning the underlying noise distribution with that encountered during pretraining. To achieve this, we introduce a learnable parameter $\phi$ into the noisy signal $y^t$, and optimize $\phi$ such that it approximates the desired deviation offset $-\epsilon^{s\to t}$.

A simple schematic diagram of the network is shown in Fig. 3.

**Self Supervised Learning** To approximate an unknown $-\epsilon^{s\to t}$ with $\phi$, we train $\phi$ ti minimize a self-supervision loss:

$$\mathcal{L}_{\text{self}} = \left\| f_\theta \left( D_1(y^t) \right) - D_2(y^t) \right\|_2^2 \tag{11}$$

where $D_1$ and $D_2$ represent two different transformations such as down sampling and average pooling; $f_\theta$ denotes the pretrained network.

Overall, our objective function to train $\phi$ becomes:

$$\phi^* = \arg\min_\phi \left\| f_{\theta\bullet} \left( D_1(y^u + \phi) \right) - D_2(y^u + \phi) \right\|_2^2 \tag{12}$$

### 4.5 FREQUENCY-AWARE ADVERSARIAL ALIGNMENT BLOCK

**Spectral Representation** To explicitly align domain statistics between the source and target distributions, we introduce a frequency-aware discriminator operating on the spectral representation of

denoised signals. Given a predicted output $\hat{x}$, we compute its 1D Fourier transform and extract the amplitude spectrum:

$$S(\hat{x}) = |\mathcal{F}(\hat{x})| \tag{13}$$

where $S(\hat{x})$ represents the spectral representation of $(\hat{x})$.

**Domain Discriminator**  The architecture of the domain discriminator $\mathcal{D}$ consists of two 1D convolutional layers with progressively increasing channel widths, followed by LeakyReLU activations and an adaptive average pooling layer that aggregates temporal features. The output is passed through a fully connected layer and a sigmoid activation to yield a scalar domain confidence score:

$$D(\hat{x}) = \text{Sigmoid}(\text{Linear}(\text{Flatten}(\text{AvgPool}(f(\hat{x}))))) \tag{14}$$

where $f(\hat{x})$) denotes the feature maps extracted by the convolutional layers.

During training, we apply a binary cross-entropy loss to supervise the discriminator with domain labels. For discriminator training, the predicted domain score $D(S(\hat{x}))$ is compared with the ground-truth domain label $d \in \{0, 1\}$ (1 for source domain, 0 for target domain):

$$\mathcal{L}_{\text{D}} = -[d \cdot \log D(S(\hat{x})) + (1 - d) \cdot \log(1 - D(S(\hat{x})))] \tag{15}$$

In parallel, the feature generator is updated adversarially using the reverse label $1 - d$, forcing the adapted prediction $\hat{x}$ to be indistinguishable from the opposite domain:

$$\mathcal{L}_{\text{adv}} = -[(1 - d) \cdot \log D(S(\hat{x}))] \tag{16}$$

This adversarial learning process encourages the model to produce frequency-domain outputs that are invariant across domains, thus facilitating improved generalization to real noisy conditions.

## 5 EXPERIMENTS

In this section, we present the experiments to address four questions:

**Q1.** How does DACDL perform in comparison to current SOTAs? (A1. See Tab. 1)

**Q2.** Is the ANL-Block genuinely model-agnostic, and what impact do they have? (A2. See Tab. 2)

**Q3.** How do different components in DACDL influence the outcomes? (A3. See Tab. 3)

**Q4.** What is the extra consumption of the ANL-Block for the existing method? (A4. See Tab. 4)

### 5.1 EXPERIMENTAL SETTINGS

**Datasets and Evaluation Metric**  Evaluation is carried out on three benchmarks in this study: (1) the Lab Dataset, in which real pump-induced noise is collected on a laboratory hydraulic-circulation setup (see Supplementary Fig. 1); (2) the Lab De-Pump Dataset, obtained by time-domain averaging of the Lab Dataset to suppress periodic pump components while primarily retaining stochastic noise and phase disturbance; and (3) the Oilfield Dataset, consisting of field recordings acquired during drilling operations in the Daqing Oilfield (the device is shown in Fig. 1). Additional dataset details are provided in the supplementary material. Notably, during episodic learning, the source domain dataset consists of Simulated Gaussian Dataset, while the target domain is the Lab Dataset, the Lab De-Pump Dataset or the Oilfield Dataset. In the target domain, only a limited number of labeled samples organized as $\mathcal{M}$ sequence are available.

To quantitatively evaluate the denoising performance of the proposed method, we employ three widely-used metrics: Mean Squared Error (MSE), Signal-to-Noise Ratio (SNR), and Structural Similarity Index Measure (SSIM).

**Implementation**  As methods tailored to pressure wave signals are scarce, representative denoising approaches from adjacent signal domains were selected as baselines. All models were trained with AdamW (learning rate $1 \times 10^{-4}$, weight decay $1 \times 10^{-4}$). For fair comparison, each architecture was first pretrained on simulated data and then fine-tuned using the $\mathcal{M}$-sequence portion of the Lab, Lab De-Pump, or Oilfield datasets. Experiments were executed on an NVIDIA RTX 4050 Laptop GPU (8 GB) with a 16-vCPU Intel i7-144650HX; the batch size was set to 2.

| Method | Parameters (M) | Oilfield Dataset | | | Lab Dataset | | | Lab De-Pump Dataset | | |
|---|---|---|---|---|---|---|---|---|---|---|
| | | MSE ↓ | SNR ↓ | SSIM ↑ | MSE ↓ | SNR ↑ | SSIM ↑ | MSE ↓ | SNR ↑ | SSIM ↑ |
| U-Net (MICCAI'15) Ronneberger et al. (2015) | 10.82 | 0.417 | 5.59 | 0.37 | 0.172 | 8.73 | 0.63 | 0.300 | 8.00 | 0.47 |
| MLWNet (CVPR'24) Gao et al. (2024b) | 10.69 | 0.525 | 4.57 | 0.21 | 0.214 | 7.59 | 0.64 | 0.365 | 6.74 | 0.43 |
| FADformer (ECCV'24) Gao et al. (2024a) | 6.96 | 1.236 | 0.84 | 0.22 | 0.115 | 10.61 | 0.73 | 0.401 | 6.20 | 0.36 |
| WaveFormer (AAAI'24) Wu et al. (2024) | 0.21 | 1.496 | 0.01 | 0.05 | 1.216 | 0.02 | 0.01 | 1.491 | 0.01 | 0.05 |
| SJDD-Net (AAAI'24) Dong et al. (2024) | **0.03** | 1.298 | 0.63 | 0.22 | 0.391 | 4.96 | 0.26 | 0.624 | 3.86 | 0.12 |
| APR-RD (AAAI'25) Kim & Cho (2025) | 0.15 | 1.27 | 0.70 | 0.13 | 0.504 | 3.86 | 0.29 | 0.327 | 2.15 | 0.14 |
| **DACDL (Ours)** | 0.61 | **0.039** | **17.64** | **0.84** | **0.054** | **14.33** | **0.84** | **0.052** | **18.00** | **0.85** |
| | | -0.378 [†] | +12.05 [†] | +0.47 [†] | -0.061 [†] | +3.72 [†] | +0.11 [†] | -0.248 [†] | +10.00 [†] | +0.38 [†] |

Table 1: Quantitative evaluation on three benchmarks. ↓ indicates that a lower value of the metric corresponds to better performance, whereas ↑ signifies that a higher value is preferable. The best-performing results are presented in bold, while the second-best results are underlined. Improvements the previous SOTA are highlighted by [†].

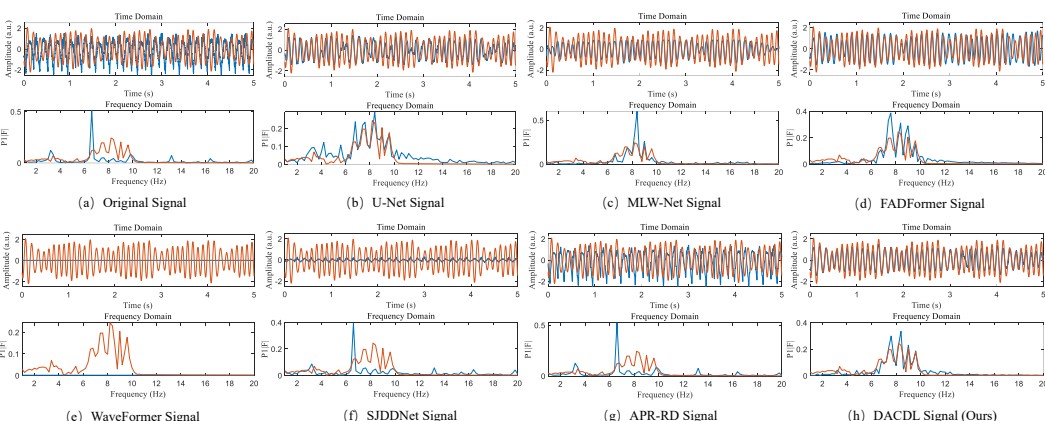

(a) Original Signal     (b) U-Net Signal     (c) MLW-Net Signal     (d) FADFormer Signal

(e) WaveFormer Signal     (f) SJDDNet Signal     (g) APR-RD Signal     (h) DACDL Signal (Ours)

Figure 4: Comparison of visualization results in the time domain and frequency domain by different methods on Lab Dataset. The orange-red curves denote the ground-truth target signals, while the blue curves represent the denoised outputs generated by each method.

## 5.2 COMPARISON WITH SOTAS

The comparative results on three datasets are presented in Tab. 1.

**On Oilfield Dataset**  As summarized in Tab. 1, our proposed method DACDL significantly outperforms all baseline approaches across all three metrics.

Specifically, DACDL achieves an MSE of 0.039, which is a dramatic reduction compared to the previous best result (0.417 by U-Net Ronneberger et al. (2015)), indicating enhanced fidelity in waveform recovery. Furthermore, it reaches an SNR of 17.64 dB, representing a gain of +12.05 dB over the next-best model, and achieves the highest SSIM of 0.84, reflecting improved structural consistency in the reconstructed signal. The improvements are indicated by [†] in the table, demonstrating the superior capability of DACDL in adapting to complex real-field signal environments.

**On Lab Dataset**  As shown in Tab. 1, our proposed method DACDL achieves the best results across all metrics, significantly outperforming existing state-of-the-art approaches. Specifically, DACDL attains the lowest MSE of 0.054, the highest SNR of 14.33 dB, and the best SSIM of 0.84. Compared to the strongest baseline, FADFormer (ECCV'25), DACDL yields a 0.061 improvement in MSE, a 3.72 dB gain in SNR, and a 0.11 increase in SSIM. These results demonstrate that incorporating our DACDL leads to substantially enhanced denoising performance, especially in retaining structural fidelity and signal quality under challenging noise conditions.

**On Lab De-Pump Dataset**  On the Lab De-Pump Dataset, DACDL again outperforms all competing baselines by a significant margin. It achieves an MSE of 0.052, an SNR of 18.00 dB, and an

SSIM of 0.85, surpassing the previous best method U-Net Ronneberger et al. (2015) by 0.248, 10.00 dB, and 0.38, respectively. Notably, the improvement in SNR indicates a major enhancement in signal recovery quality, while the substantial gain in SSIM reflects better preservation of the original signal structure.

Parts of visualization results on Lab Dataset are displayed in Fig. 4.

| Method | | Lab Dataset | | |
|--------|--|-------------|--|--|
| | | MSE ↓ | SNR ↓ | SSIM ↑ |
| U-Net (MICCAI'15) | Reproduced | 0.172 | 8.73 | 0.63 |
| | Reproduced + ANL-Block | **0.124** | **10.24** | **0.76** |
| | | -0.048 [†] | +1.51 [†] | +0.13 [†] |
| MLWNet (CVPR'24) | Reproduced | 0.214 | 7.59 | 0.64 |
| | Reproduced + ANL-Block | **0.154** | **9.98** | **0.78** |
| | | -0.06 [†] | +2.39 [†] | +0.14 [†] |
| FADformer (ECCV'24) | Reproduced | 0.115 | 10.61 | 0.73 |
| | Reproduced + ANL-Block | **0.089** | **11.87** | **0.79** |
| | | -0.026 [†] | +1.26 [†] | +0.06 [†] |

Table 2: Evaluation of the model-agnostic functionality of the ANL-Block. *Reproduced* denotes our reproduction of the baseline results. *+ ANL* illustrates the enhanced baseline results on the Pump Nosie Dataset with the addition of the ANL-Block. Improvements over the reproduction are highlighted by [†].

## 5.3 MODEL-AGNOSTIC FUNCTIONALITY OF THE ANL-BLOCK

As previously discussed, the proposed ANL-Block is designed to be model-agnostic, allowing seamless integration with a wide range of existing methods. To substantiate this claim and further demonstrate the effectiveness of our approach, we selected three representative open-source models as baselines. These baseline methods were first faithfully reproduced to ensure experimental consistency, after which the ANL-Block was incorporated into each architecture. The corresponding results are presented in Tab. 2.

The findings indicate that, in the vast majority of cases, integrating the ANL-Block yields noticeable performance improvements over the original models, with gains ranging from 0.02 to 2.39. These results underscore the flexibility and practical value of the ANL-Block1 as a plug-and-play module.

## 5.4 ABLATION STUDY

**Overall** To comprehensively evaluate the contributions of each proposed component within the DACDL architecture, we conduct a series of ablation studies on the Lab Dataset, as presented in Tab. 3. As shown in row ①, the baseline model trained only on simulated data without any enhancement modules, yields modest performance with an MSE of 0.339, SNR of 6.84 dB, and SSIM of 0.44. When progressively incorporating the EL-Framework, ANL-Block, and FAA-Block, we observe consistent improvements across all evaluation metrics. Notably, the complete model (row ⑧) that integrates all three components achieves the best performance with an MSE of 0.054, SNR of 14.33 dB, and SSIM of 0.84, demonstrating the synergistic effectiveness of our design.

**Episode Learning Framework** The EL-Framework plays a pivotal role in bridging the domain gap between simulated and real noise conditions. As shown in row ②, enabling the EL-Framework alone significantly improves performance over the baseline, reducing the MSE from 0.339 to 0.251 and boosting the SNR from 6.84 dB to 10.84 dB. This confirms that the EL-Framework successfully facilitates domain adaptation by learning a more robust initialization from simulated data while effectively leveraging a small amount of real data through latent distribution alignment.

**Adaptive Noise Learning Block** The Adaptive Noise Learning Block (ANL-Block) introduces a self-guided mechanism for modeling input noise perturbations. When used independently, as presented in row ③, the ANL-Block improves performance compared to the baseline, yielding an MSE of 0.287, SNR of 9.78dB, and SSIM of 0.58. Furthermore, when jointly used with the EL-Framework, as illustrated in row ⑤, the model attains stronger performance with an MSE of 0.128

and SNR of 13.09 dB. These results demonstrate the ANL-Block's capacity to extract domain-aware noise characteristics that enhance the generalizability of the denoising network.

**Frequency-aware Adversarial Alignment Block**   The Frequency-aware Adversarial Alignment Block (FAA-Block) is designed to emphasize informative spectral bands relevant to noise suppression. As shown in row ④, the inclusion of FAA alone achieves improved performance over the baseline (MSE: 0.309, SNR: 8.35 dB, SSIM: 0.51). When combined with the EL-Framework (see row ⑥) or the ANL-Block (see row ⑦), the model further benefits from multi-scale spectral guidance, leading to notable gains in denoising quality. Ultimately, the combination of all three modules, as reported in row ⑧, confirms that FAA-Block complements EL-Framework and ANL-Block in a synergistic manner.

| | EL-Framework | ANL-Block | FAA-Block | Lab Dataset | | |
|---|---|---|---|---|---|---|
| | | | | MSE ↓ | SNR ↑ | SSIM ↑ |
| ① | | | | 0.339 | 6.84 | 0.44 |
| ② | ✓ | | | 0.251 | 10.84 | 0.69 |
| ③ | | ✓ | | 0.287 | 9.78 | 0.58 |
| ④ | | | ✓ | 0.309 | 8.35 | 0.51 |
| ⑤ | ✓ | ✓ | | 0.128 | 13.09 | 0.76 |
| ⑥ | ✓ | | ✓ | 0.154 | 11.64 | 0.71 |
| ⑦ | | ✓ | ✓ | 0.143 | 12.78 | 0.73 |
| ⑧ | ✓ | ✓ | ✓ | **0.054** | **14.33** | **0.84** |

Table 3: Ablation studies of different components in DACDL on Lab Dataset. ① denotes the pre-trained model.

## 5.5 EXTRA CONSUMPTION

As mentioned earlier, the proposed ANL-Block is a model-agnostic module that can be easily integrated into most existing architectures. Its integration introduces only minimal computational overhead. As reported in Tab. 4, taking U-Net Ronneberger et al. (2015) as an example, the original model has 10.82M parameters and requires 65.17 seconds for execution. With the ANL-Block incorporated, the parameter count increases slightly to 10.88M, and the runtime extends to 67.23 seconds. This corresponds to an overhead of just 0.06M parameters and 2.06 seconds in runtime. Given its minimal computational overhead and the substantial performance gains it enables, the ANL-Block offers a highly efficient and practically valuable enhancement to existing models.

| Method | Param (M) | Training Time | Lab Dataset | | |
|---|---|---|---|---|---|
| | | | MSE ↓ | SNR ↑ | SSIM ↑ |
| U-Net | 10.82 | 65.17s | 0.172 | 8.73 | 0.63 |
| U-Net + ANL | 10.88 | 67.23s | 0.124 | 10.24 | 0.76 |

Table 4: Experiments of consumption with one 4050 Laptop GPU on Lab Dataset. The training time specified covers pre-training time and fine-tuning time.

## 6 CONCLUSION

In this paper, we presented DACDL, a Deep Adaptive Cross Domain Learning framework tailored for robust signal recovery in real-world mud pulse telemetry applications. The framework integrates three key innovations: an Episode Learning Framework (EL-Framework) for cross-domain adaptation, a plug-and-play Adaptive Noise Learning Block (ANL-Block) for input-level noise transformation, and a Frequency-aware Adversarial Alignment Block (FAA-Block) for spectral domain alignment. Experiments across three datasets have demonstrated the effectiveness of the proposed method and the value of each component.

## 7 REPRODUCIBILITY STATEMENT

Our method is reproducible. The model architecture is illustrated in Fig. 2, and the hyperparameters as well as training details are described in Sec. 5. Fig. 5 and Fig. 6 in the appendix demonstrate the hydraulic circulation setup utilized for Lab Dataset collection and the signal characteristics of the datasets used, respectively. In addition, the source code will be made publicly available in the future.

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

## A  RELEVANT CONCEPTIONS

In this section, a detailed introduction to the time-domain averaging method mentioned in the main text is provided below.

### A.1  TIME-DOMAIN AVERAGING

Time-Domain Averaging (TDA) McFadden (1987); Braun (2011) is a classical signal processing technique widely used to suppress periodic or quasi-periodic noise and enhance the underlying signal features. In the context of mud pulse telemetry or other pressure-based transmission systems, TDA operates by segmenting the raw signal into multiple cycles aligned to a common reference, and subsequently averaging these segments point-by-point across time. This process effectively reinforces the consistent components of the signal while attenuating uncorrelated noise and random disturbances, such as Gaussian noise and non-periodic artifacts. The assumption underlying TDA is that the desired signal exhibits strong periodicity or repeatable patterns, whereas the noise is stochastic and varies across cycles. As a result, TDA serves as a powerful preprocessing step for improving signal quality and reliability prior to further analysis, decoding, or denoising via learning-based models.

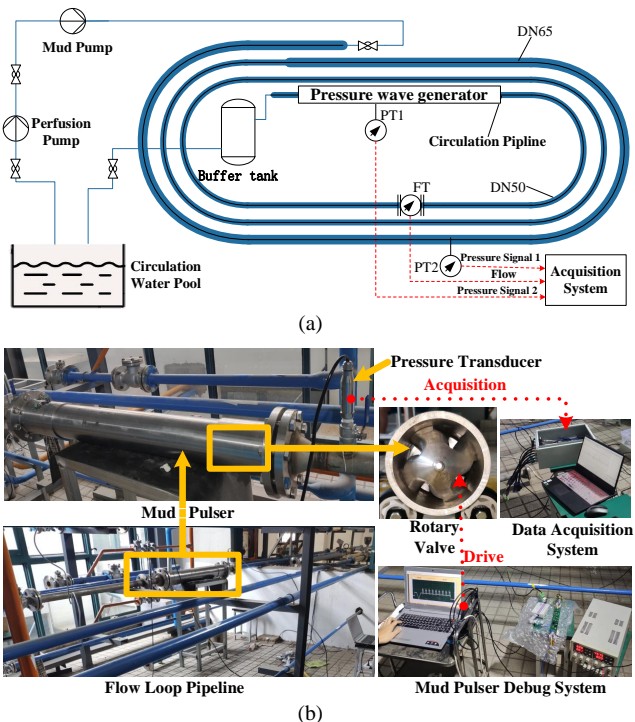

(a)

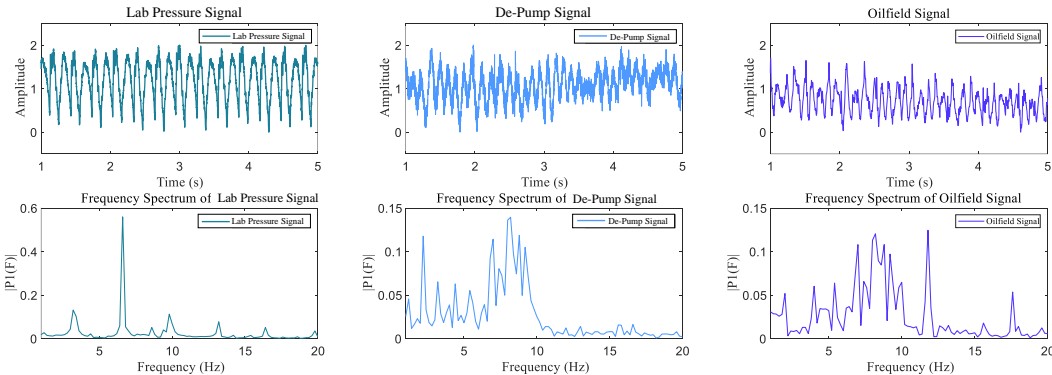

(b)

Figure 5: Hydraulic Circulation Setup. The system mainly consists of a circulating water pool, a perfusion pump, a mud pump, a buffer tank, a pressure wave generator, a hydraulic circulation pipeline, pressure and flow sensors (PT1, PT2, FT) and a data acquisition system.

Figure 6: Illustration of representative time-domain waveforms (top row) and frequency spectra (bottom row) for three types of signals: Lab Pressure Signal, De-Pump Signal, and Oilfield Signal.

## B   SUPPLEMENTED EXPERIMENTS

### B.1   DATASETS DESCRIPTION

This section will provide a detailed introduction to the characteristics of each dataset.

**Lab Dataset**    The Lab Dataset was collected using the experimental setup shown in Fig. 5. The pressure source comprised two centrifugal pumps that provided a stable flow. The loop pipeline was composed of two sections of stainless-steel pipes with lengths of 72.7 m and 82.1 m and diameters of 60 mm and 50 mm, respectively. Pressure transmitters were installed at the ends of the mud pulser and upstream of the pipeline. The rotary valve continuous wave mud pulser that we developed can

generate mud pulses of up to 30 Hz. Since drilling mud or water did not change the propagation law of pressure waves, the circulating fluid was replaced with water according to laboratory conditions.

Pressure transmitters were installed at two key locations: one at the outlet of the mud pulser (PT2), serving as the reference measurement minimally affected by transmission noise, and one at the end of the pipeline (PT1), recording the received signal after propagation through the entire loop, thus containing accumulated noise from pumps, turbulence, and environmental disturbances. All signals were synchronously recorded using an acquisition system. As drilling mud and water exhibit similar pressure wave propagation properties, water was used as the circulating fluid for convenience under laboratory conditions. The resulting dataset provides paired noisy and clean pressure waveforms under fixed operating conditions, enabling realistic evaluation of denoising and signal reconstruction algorithms in mud pulse telemetry scenarios.

**Lab De-Pump Dataset** The Lab De-Pump Dataset was obtained after performing the TDA method on the Pump Noise Dataset. Despite the denoising process, it still contains residual Gaussian noise and phase distortion.

**Oilfield Dataset** The Oilfield Dataset was collected from an operational mud pulse telemetry (MPT) system deployed in a field drilling environment, as illustrated in Fig. 1 (a). In this system, downhole pressure pulses are generated at the drill bit by a signal generator and transmitted upward through the drilling fluid column to the surface. The pressure signals propagate along the mud channel, where they are subjected to complex noise sources and interference arising from turbulent fluid flow, mechanical vibrations, and interactions with the drilling assembly. A mud pump circulates the drilling fluid, further introducing structured and unstructured noise components. Pressure transducers installed at the surface continuously record the received pressure waveforms, which are then transmitted to the ground processing unit for further analysis.

**Signal Characteristic Description** Fig. 6 presents representative time-domain waveforms and frequency spectra from three datasets. The Lab Pressure Signal is characterized by strong periodic noise components that dominate the waveform and obscure the underlying signal structure. In the frequency domain, this is reflected by pronounced and narrow spectral peaks, indicative of stable and repetitive interference. The phase information in this signal is largely ambiguous due to the overwhelming presence of periodic noise, which renders accurate interpretation of the original waveform difficult.

The De-Pump Signal created by time-domain averaging of the Lab Dataset presents a comparatively cleaner waveform in which the periodic components have been significantly suppressed. Its frequency spectrum shows a more continuous distribution of energy with reduced peak dominance, suggesting the presence of background Gaussian noise rather than structured interference. However, residual phase distortion and low-amplitude fluctuations remain, indicating that the signal is not entirely free from noise contamination.

The Oilfield Signal collected from actual oilfield operations exhibits complex structure. It contains irregular, non-repetitive interference patterns and displays substantial spectral spreading in the frequency domain. The energy is dispersed across a wider range of frequencies, and the signal exhibits severe phase distortion and high variability. These characteristics reflect the diverse and unpredictable noise conditions encountered in real-world downhole environments.

The three datasets collectively encompass a range of noise complexities, from structured periodic interference to unstructured and irregular noise, thereby providing a comprehensive basis for evaluating signal analysis and recovery methods under increasingly challenging conditions.

**The Reason for not Choosing Public Datasets** In this study, we do not use any publicly available mud pulse telemetry datasets for evaluation. This decision is primarily due to the following reasons:

Lack of Public Benchmarks: To the best of our knowledge, there are currently no publicly accessible, standardized datasets for mud pulse telemetry signal recovery. Most existing datasets are proprietary and have not been released for open research.

Confidentiality and Intellectual Property: Mud pulse telemetry data are often considered sensitive due to commercial interests and intellectual property concerns. The field data used in this work were

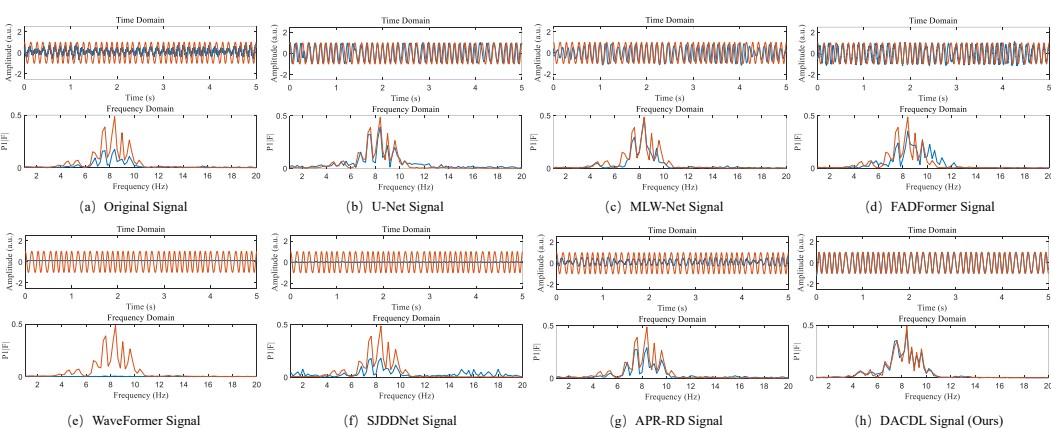

Figure 7: Comparison of visualization results in the time domain and frequency domain by different methods on Real Gaussian Dataset. The orange-red curves denote the ground-truth target signals, while the blue curves represent the denoised outputs generated by each method.

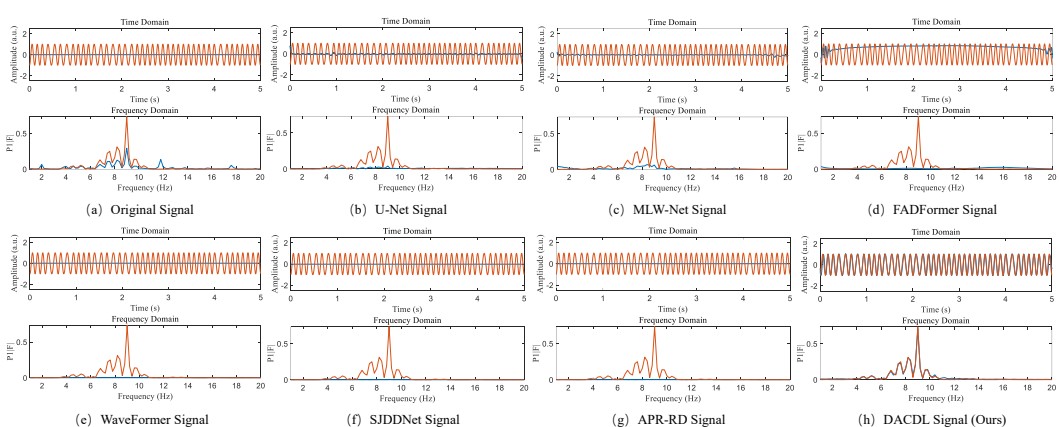

Figure 8: Comparison of visualization results in the time domain and frequency domain by different methods on Real Oilfield Dataset. The orange-red curves denote the ground-truth target signals, while the blue curves represent the denoised outputs generated by each method.

obtained in collaboration with industrial partners under non-disclosure agreements, which preclude public sharing.

Practical Relevance: The datasets collected in this study closely reflect the complexities and noise characteristics encountered in real-world drilling operations, providing a realistic and challenging testbed for evaluating the proposed methods.

As a result, all experiments in this paper are conducted on our in-house datasets, which are described in detail in the above section. We encourage future work to promote the development and open sharing of standardized benchmarks for mud pulse telemetry research.

### B.2 MORE VISUALIZATION RESULTS

We perform additional visualizations on the other two datasets, as illustrated in Fig. 7 and Fig. 8.

## C THE USE OF LARGE LANGUAGE MODELS (LLMS)

We used a large language model (LLM) for language polishing of this manuscript (grammar correction, wording clarity, and minor stylistic edits) and for coding assistance (e.g., clarifying error messages, suggesting syntax fixes, minor refactoring, and correcting small implementation mistakes). These supports made our research processsmoother and more effective.

