# OpenReview forum: "Deep Adaptive  Cross Domain  Learning for Continuous Pressure Wave Signal Recovery"
_ICLR.cc/2026/Conference — Submitted to ICLR 2026_

### Official Review · Reviewer_k8S1 · 2025-10-20

**Soundness:** 2
**Presentation:** 1
**Contribution:** 2
**Rating:** 2
**Confidence:** 3

**Summary:**

This work proposes an episodic learning framework, which learns the shift from the simulated noise domain to the real pump noise domain, which alleviates scarcity of annotated noise data in mud pulse telemetry problem. Through the episodic learning framework, this work designs a lightweight Adaptive Noise Learning Block and A Frequency-aware Adversarial Alignment Block to reconstruct the noisy signal. Abundant experiments in datasets collected from the real world demonstrate that the proposed method achieves great performance.

**Strengths:**

1.	This work can achieve joint learning from simulated and real data by episodic learning to alleviate the issue of label data sparsity.
2.	This work enhances the performance of signal reconstruction by enforcing spectral invariance through adversarial training in the Fourier domain.

**Weaknesses:**

1.	The organization of the article is rather confused, making it difficult to quickly grasp the methodology pipeline.
2.	The abbreviation FSL appears on line 125, but it is not explained until line 208.
3.	Lack of research into relevant literature for the traditional signal denoise methods and episodic learning.
4.	The meaning of S(u) in eq 3 is not specified. If it denotes $(\mathcal{F}s)(u)$, please specify this explicitly in the main text.
5.	WaveU-Net is not cited and introduced.
6.	There are some typos in the main text, e.g., the "ti" at line 255, the "f(x))" at line 281.
7.	The meaning of y^u in eq 12 is not specified.
8.	S appears at eq 3 and eq 13, and their meanings seem to be different.
9.	Lack of explanation regarding the model structure.
10.	Lack of description of the dataset size.
11.	The writing of this article is puzzling and the specific architectural design of the network is unknown.
12.	The introduction of existing deep learning methods and traditional approaches is insufficient, making it difficult to fully understand the contribution of this article.
13.	WaveU-Net and LEDNet do not have citations. No explanation is given as to why it is reasonable.
14.	Some meanings of symbols have no explanations, e.g., S(u) in equation (3), y^u in equation (12), and some symbols are in disarray, e.g., S in equation (13) and in equation (3).

**Questions:**

1.	What does "feature generator" at line 287 represent? Is it the same as "feature extraction" in section 4.2?
2.	What is the setting of the transformations at eq 12? Is it randomly assigned? Or does it have a specific practical meaning?
3.	In ADL-block, is \phi (eq 12) directly obtained for each sample through optimization methods? Or is a Phi-Net learned to map from y to \phi? This is not specified in the text or training objective.
4.	The method involves multiple modules (Phi-Net, Feature Extractor, and Classifier) and multiple training objectives (L_self, L_sft, L_D, L_adv), but the training pipeline remains unclear. Are they trained together or separately? If separately, what is the training sequence?
5.	The methods compared in Table 1 are mostly techniques for 2-D image signal denoising, whereas the signals in this work are 1-D. For the completeness of this work, it is necessary to compare with the latest 1-D signal reconstruction methods.
6.	How does DACDL compare against classic unsupervised domain adaptation methods?
7.	Why select 500 length window on the line 132?
8.	I don't understand why a sampling and average pooling operation needs to be performed on the signal input at line 261?
9.	The specific size of the test dataset?
10.	What is the input of feature extractor?
11.	What is the architecture of the pre-trained model for Phi-Net? How to pretrain this model?

**Details Of Ethics Concerns:**

"In prior work, we introduced WaveU-Net" appears at line 184, which is the first mention of "WaveU-Net". I'm concerned that this violates the double-blind rule.

---

### Official Review · Reviewer_Dqbx · 2025-10-29

**Soundness:** 3
**Presentation:** 2
**Contribution:** 2
**Rating:** 4
**Confidence:** 3

**Summary:**

This paper proposes DACDL, a Deep Adaptive Cross Domain Learning framework for recovering continuous pressure wave signals in Mud Pulse Telemetry (MPT) systems. The approach introduces three components:
(1) an Episodic Learning Framework (EL-Framework) for alternating meta-learning between simulated and real domains;
(2) a model-agnostic Adaptive Noise Learning Block (ANL-Block) that injects learnable perturbations to align noise distributions;
(3) a Frequency-Aware Adversarial Alignment Block (FAA-Block) that aligns spectral characteristics across domains.
Experiments on three proprietary datasets (Lab, Lab De-Pump, and Oilfield) show significant improvements compared with several signal denoising baselines.

**Strengths:**

1. The combination of episodic meta-learning, adaptive noise perturbation, and frequency-domain alignment is conceptually coherent and well-implemented.
2. The reported performance gains across all metrics (MSE, SNR, SSIM) and datasets are substantial.
3. The ANL-Block’s plug-and-play design demonstrates some general utility.
4. The methodology and ablation studies are clearly presented, and the reproducibility statement is detailed.

**Weaknesses:**

1. The paper entirely relies on proprietary, non-public datasets from industrial partners. As a result, the experimental results cannot be independently verified by the community, which substantially weakens the empirical credibility and reproducibility expected for ICLR-level publications.
2. The proposed framework is tailored specifically to mud pulse telemetry, which is an extremely narrow and specialized field in petroleum engineering. It is unclear whether DACDL can generalize to other signal recovery or cross-domain learning problems, such as audio denoising, seismic inversion, or sensor signal adaptation. Demonstrating results on a more general or publicly available dataset would be essential to justify broader impact.
3. While technically well-executed, the paper’s core contribution lies in a highly application-specific engineering problem rather than a fundamental advance in representation learning, optimization, or general domain adaptation theory, which are the main focus areas of ICLR.
4. The ANL-Block is claimed to be “model-agnostic,” but no evidence is shown on tasks beyond MPT signal denoising. Without cross-domain experiments in other modalities, this claim remains unconvincing.

**Questions:**

1. Can DACDL be applied to more general domains (e.g., speech or vibration signal recovery)? If so, please show at least one cross-domain transfer experiment.
2. Are the datasets or synthetic simulators intended to be released for reproducibility?
3. How sensitive is the performance to the number of labeled samples in the target domain?
4. Could the improvements stem mainly from dataset bias rather than genuine domain adaptation?

---

### Official Review · Reviewer_WaSH · 2025-10-31

**Soundness:** 2
**Presentation:** 1
**Contribution:** 2
**Rating:** 2
**Confidence:** 3

**Summary:**

This paper proposes DACDL, a Deep Adaptive Cross-Domain Learning framework for denoising continuous pressure-wave signals in mud-pulse telemetry (MPT) systems. The goal is to improve model generalization from synthetic (Gaussian) to real (pump and field) noise environments. Experiments on three in-house datasets (Lab, De-Pump, Oilfield) show that DACDL improves SNR and SSIM over prior denoising baselines such as U-Net, FADFormer, and MLWNet.

**Strengths:**

1. The paper tackles a realistic and underexplored industrial problem.
2. Showing performance improvement over selected baseline models.
3. The proposed ANL-Block is designed to be model-agnostic, allowing seam-
less integration with a wide range of existing methods.

**Weaknesses:**

1. In Section 4.2 (Page 4, Line 184), the authors write “In prior work, we introduced WaveU-Net, a wavelet-enhanced U-Net that achieved strong results in signal restoration.” with first-person phrasing (“we introduced”). This may reveal the authors' identity and could therefore conflict with the double-blind policy.

2. Lack of fair comparisons. The paper does not include comparisons with existing domain-adaptation or transfer-learning methods, limiting the validity of the claims.

3. The meaning of model-level versus input-level adaptation is not formally defined.

4. The paper lacks a section discussing related works, e.g., prior domain-adaptation, existing denoising methods, or cross-domain learning methods.

5. The writing needs to be improved. For example: 1) Figure 1(b) contains little information, lacking legend and clarity. 2) Multiple typos and formatting issues appear (e.g., page 3 line 128, Eq.3 incomplete notation definition, page 5 line 256). 3) $y^{s\to t}$ (page 5 line 243) used without definition. 4) Figure 3 is visually cluttered and lacks clear labeling.

6. The proposed episode-based learning setup is quite unclear and the role of support and query sets is not explained.

**Questions:**

1. Wwhy minimizing feature differences between two augmentations $(D_1, D_2)$ relates to correcting the domain deviation. In standard self-supervised learning, this kind of loss enforces invariance under data augmentations. Here, it's proposed to “align” distributions, but that connection isn't justified.

2. $\epsilon^{s\to t}$ should be treated as a distribution rather than a single deterministic offset.

3. The motivation for conducting adversarial alignment in the frequency domain rather than the time or latent domain is not theoretically justified or empirically analyzed.

---

### Meta-Review · Area_Chair_3Sdp · 2025-12-27

**Summary:**

Reviewers have pointed out relevant concerns regarding broader impact for the ICLR community, but also regarding writing and the limited positioning of the work with respect to the state of the art. For these reasons the work cannot be endorsed for publication at ICLR 2026.

**Reviewer Concerns:**

None. Rebuttal was not submitted.

**Reviewer Scores:**

Reviewer WaSH: 2
Reviewer Dqbx: 4
Reviewer k8S1: 2

---

### Decision · Program_Chairs · 2026-01-26

Reject